# Gold Clusters Immobilized by Post-Synthesis Methods on Thiol-Containing SBA-15 Mesoporous Materials for the Aerobic Oxidation of Cyclohexene: Influence of Light and Hydroperoxide

**Rafael Delgado** [1,2], **Carlos Márquez-Álvarez** [1] , **Álvaro Mayoral** [3,4,5] , **Ramón de la Serna** [1] , **Javier Agúndez** [1] and **Joaquín Pérez-Pariente** [1,*]

1 Instituto de Catálisis y Petroleoquímica (CSIC), 28049 Madrid, Spain
2 Instituto de Nanociencia, Nanotecnología y Materiales Moleculares (INAMOL),
  Universidad de Castilla-La Mancha, 13001 Ciudad Real, Spain
3 Instituto de Nanociencia y Materiales de Aragón (CSIC-UNIZAR), 50009 Zaragoza, Spain
4 Laboratorio de Microscopías Avanzadas (LMA), Universidad de Zaragoza, 50009 Zaragoza, Spain
5 Shanghai Key Laboratory of High-Resolution Electron Microscopy, ShanghaiTech University,
  Shanghai 201210, China
* Correspondence: jperez@icp.csic.es

**Abstract:** Gold nanospecies produced by a historically inspired two-liquid phase system were immobilized on plate-like mesoporous silica, SBA-15, functionalized with mercaptopropyl groups by a post-synthesis method, and the resulting materials were tested in the oxidation of cyclohexene with molecular oxygen at atmospheric pressure. The main purpose of this approach was to compare the physicochemical properties and catalytic performance of these materials with those of previously reported related materials functionalized by in situ methods during synthesis. In addition, catalytic tests under ambient lighting and darkness and also in the presence and absence of the initiator *tert*-butyl hydroperoxide (TBHP) were carried out. The samples were characterized by chemical analysis, $N_2$ adsorption/desorption, TGA, SEM, HRTEM, UV-vis spectroscopy and XPS. Gold nanoclusters and isolated gold atoms but no AuNPs were found in the catalysts (0.31–2.69 wt.% of gold). The XPS shows that nearly 60% of the -SH groups (1.33 wt.% of S) were oxidized to sulphonic groups upon gold immobilization. The AuNCs and isolated gold atoms evolved in the the reaction medium to form AuNPs. The activity of the samples was lower than that of the catalysts supported on related S-bearing SBA-15 functionalized in situ, which was attributed to their different Au/S ratios, which in turn regulated the evolutionary process of the gold species during the reaction. The catalysts turned out to be inactive in darkness, which evidences that the cyclohexene oxidation carried out at ambient illumination is actually photocatalyzed by the AuNPs formed in situ during the reaction. The TBHP initiator is required to obtain the activity in order to counteract the inhibitors of cyclohexene auto-oxidation present in the commercial reagent. On the other hand, no major differences in the selectivity among the different catalysts and reactions were observed, with 2-cyclohexen-1-one and 2-cyclohexen-1-ol resulting from the allylic oxidation as main products (selectivity of (one + ol) ~80% at a conversion $\geq$ 35%; one/ol~2).

**Keywords:** gold nanoclusters; catalysis; cyclohexene; oxidation; SBA-15; functionalization; photocatalysis; mercaptopropyl; hydroperoxide



## 1. Introduction

Studies on gold catalysis are deserving of continuous and increasing interest since the seminal papers were published in the mid-1980s by Haruta and co., who reported that supported gold nanoparticles (AuNPs) are active in the oxidation of CO at low temperature [1], and by Hutchings on the hydrochlorination of ethylene [2] (it is, however, of note that prior

to these works, a number of reports on gold catalysis had already been published, and many more were covered in patent literature [3]). The field is flourishing in diversity, and new avenues for the synthesis of gold nanoentities, particularly gold nanoclusters (AuNCs) (typically gold particles with a size below 2 nm [4], encompassing both disordered and well-defined nanostructures) and their corresponding catalytic behavior in a large variety of different chemical processes [5–7] are subjected to increasing exploration [8–11]. For their use in catalysis, these gold nanoentities are generally supported on suitable solids to produce heterogeneous catalysts, endowed with their well-known advantages, particularly the role of the support in preventing the agglomeration of the metal particles, which, if uncontrolled, would ultimately lead to poor catalytic performance owing to the rapid decrease in the surface gold atoms available for the activation of the incoming molecules as the particle size increases. In addition to this stabilizing role, the support can also strongly influence the specific activity of the gold entities through the development of specific chemical interactions, which can modulate the metal activity [12].

It is within these efforts to enlarge the scientific horizons of gold catalysis that, some years ago, we started a systematic survey of the historical literature concerning gold chemistry. The scientific rationale for this apparently odd approach relies on the fact that many chemical processes carried out in the pre-Lavoisier era (if we can use this well-known eighteenth century French chemist as a historical reference of the sign of the beginning of modern chemistry) involved methodologies that have since been neglected in favor of eventually more efficient routes, not to speak of the fact that the theoretical framework that animated and inspired past chemical experiences is completely alien to modern science. In consequence, there is a huge mass of historical chemical literature waiting to be examined through modern eyes. Concerning gold in particular, this metal has long since and up to the eighteenth century been used as an ingredient in pharmaceutical preparations, because its chemical incorruptibility was considered a sign of perfection, a property that was believed to endow it with the capability of healing the most severe ailments, provided that the gold could be assimilated by the human body. In pursuing this objective, a large variety of different methods were developed to prepare drinkable gold solutions, which were known as potable gold. In the process of surveying the abundant available literature on this subject, we paid attention to a procedure reported in the 1756 posthumous edition of *Cours de Chymie*, a renowned chemical book authored by the well-known French chemist and apothecary N. Lémery [13]. The editor of this book acknowledges that the recipe was actually due not to Lémery himself but to Mademoiselle de Grimaldi, a Savoy-born woman who together with her father developed empirical medicines in their homeland in the first decades of the eighteenth century. Grimaldi's recipe is surprisingly simple, a nice example of past chemical ingenuity. This is a two-liquid phase procedure that involves a gold solution in aqua regia and rosemary oil. The reproduction of this recipe has shown the presence of gold nanoparticles, clusters and even isolated atoms in the oily phase [14]. This discovery prompted us to improve the recipe to increase the yield of AuNCs and to immobilize them in a suitable support to develop heterogeneous catalysts. Both objectives were successfully achieved, and the AuNCs supported on the mesoporous SBA-15 functionalized with 3-mercaptopropyl groups are active catalysts for the aerobic oxidation of cyclohexene under atmospheric pressure and moderate temperature (65 °C), as well as selective toward the allylic ring oxidation [15]. Catalysts prepared by replacing rosemary oil with eucalyptus oil show similar catalytic features [16]. It has also been found that the functionalization of that mesoporous support with 3-aminopropyl groups greatly enhances the specific catalytic activity (TON) of gold [17]. These studies were carried out with SBA-15 of a conventional rod-like morphology, where the mesoporous channels run along the main axis of the elongated particles. This long diffusional pathway could eventually be detrimental for catalytic activity. Indeed, it has been shown that this is the case, as the activity increases if plate-like SBA-15 is used, where the channels now run along the short plate axis [18].

All of these studies were carried out with SBA-15 functionalized by in situ methodologies, where the functional alkoxysilane precursors are added to the synthesis gels. However, it was found that the presence of the functional group somehow distorted the plate morphology of the resulting SBA-15. Therefore, it would be interesting to explore the behavior of this mesoporous material functionalized using post-synthesis methods, which would not only preserve the pristine SBA-15 plate-like morphology but would also eventually lead to an arrangement of the S-bearing functional groups different than those resulting from in situ methods that might affect the catalytic performance of the resulting materials. This was one of the objectives of this work. In addition, other aspects were also considered here. Catalytic experiments at atmospheric pressure are commonly carried out in glass flasks, which are usually run unprotected against environmental light. This is, indeed, the system we used in this and the other previous works listed in the references. Owing to the well-known light-activated plasmonic catalysis by AuNPs [19,20] and also the photocatalytic activity displayed by AuNCs [21,22], it would be interesting to determine if our Au/SBA-15 system is eventually affected by environmental illumination conditions, which encompass a wide range of wavelengths ($\lambda > 400$ nm) prevalent during the catalytic experiments. This work is a preliminary assessment of this aspect, which might eventually lead to more detailed and focused on-purpose studies on the photocatalytic activity of these Au/SBA-15 systems. In addition, we also explored the influence of *tert*-butyl hydroperoxide (TBHP) used as an initiator in the reaction by carrying out experiments in the absence of this reagent.

## 2. Materials and Methods

### 2.1. Functionalization of Plate-like SBA-15 with 3-Mercaptopropyltrimethoxysilane (MPTMS)

Pure silica SBA-15 having plate-like morphology was synthesized from gels with the chemical composition (save for the absence of MPTMS) and according to the procedure described in [18]. The as-made samples (approximately 2 g) were calcined in a tubular quartz reactor, first in a flow of $N_2$ (100 mL/min) from RT until 350 °C, where it remained for 3 h, and then it was heated up to 500 °C for 1 h. After this, the nitrogen was replaced by air (100 mL/min) and kept at that T for 5 h. The heating rate was 2 °C/min. The calcined sample was denoted as SBA-15c. To functionalize the material with MPTMS, 3 g of calcined SBA-15 was placed in a two-necked, round-bottom 250 mL flask. To remove the adsorbed water, the flask was heated in a silicon oil bath at 80 °C under vacuum for 16 h. The system was then purged with $N_2$, 16 mL of dry toluene was added, and the mixture was stirred to obtain the silica material well dispersed in the toluene. Then, 1.76 mL of MPTMS was added. While flowing $N_2$ through the system, a condenser was adapted to the flask, which was then heated at 130 °C for 24 h under continued stirring. The solid was washed first with dry toluene and then with acetone, filtered and dried at 30 °C. A total of 2.76 g of functionalized sample was recovered. The sample was denoted as SBA-SH.

### 2.2. Synthesis of the Au NCs

The procedure previously described in [15,17,18] was followed. A gold lump of 0.1419 g (Johnson-Matthey, London, UK, 99.99%) was dissolved in 45.41 g of aqua regia (prepared by dissolving ammonium chloride (Sigma Aldrich, St. Louis, MO, USA, >98%) in nitric acid (Panreac, Barcelona, Spain, 65%) in a 4:1, *w/w* ratio) under gentle stirring while heating in a sand bath at 40 °C. Once cooled, the golden-yellow solution was placed in a 100 mL decanting funnel, and 22.7 g of rosemary essential oil (supplied by The Integral Barn, main components: 1,8-cineole (24.9%), alpha-pinene (21.9%) and camphor (20.9%); its full chemical composition is reported in [18]) was added. The system was left undisturbed (not stirred), and aliquots of the upper layer consisting of the rosemary oil were taken at selected time intervals to prepare the Au/SBA-15 materials, as explained below.

### 2.3. Immobilization of the Au NCs on the Functionalized SBA-15

Aliquots of 5.25 mL of the organic phase were taken after three and eight days of the addition of the rosemary oil and diluted with 26.25 mL of ethanol. A total of 0.700 g of

the functionalized SBA-15 material was then added, and the mixture was stirred for 4 h. After, the solid was recovered by centrifugation, washed with ethanol and dried at 30 °C overnight. The corresponding samples were denoted as 3d-Au and 8d-Au, where the first figure indicates the time at which the rosemary oil aliquot was taken. In addition, one aliquot of 10.5 mL of the organic phase taken after three days was also used to immobilize the gold, and the resulting sample was denoted as 3d-Aux2.

### 2.4. Catalytic Tests

The oxidation of cyclohexene with molecular oxygen was carried out in a 50 mL four-necked, round-bottom flask, provided with a condenser through which water at 5 °C was circulated to minimize evaporation. The flask was immersed in a silicon oil bath to maintain a reaction temperature of 65 °C, measured with a thermometer inserted through one of the necks down into the reaction mixture, of a composition adapted from that reported in [23], which was composed of the following reagents: 5.6770 g of cyclohexene (0.0676 mol, Sigma-Aldrich, 99%, stabilized with butylhydroxytoluene (BHT)), 0.5677 g of octane (10 wt.% referred to cyclohexene; Sigma-Aldrich, >99%), 0.2839 g of a *tert*-butyl hydroperoxide solution (TBHP, 5 wt.% referred to cyclohexene, ~5.5 M in decane, Sigma-Aldrich), 4.2578 of toluene (75 wt.% of the cyclohexene, Panreac, >99.5%) and 0.070 g of catalyst. $O_2$ (1.8 mL/min) was bubbled through the stirred reaction mixture. Prior to the addition of the reagents, the catalysts were heated at 100 °C for 1 h in the reaction flask, which was connected to a tube containing molecular sieve 5A to remove the traces of water present in the catalysts. After this treatment, the flask was cooled down to the reaction temperature and the reagents were added. The reaction time was counted from the moment in which the reaction temperature reached 65 °C. Aliquots of 0.2 mL of the reaction mixture were taken at selected time intervals, filtered through a 4 mm PTFE filter of 0.45 μm and analyzed by GC in a Varian CP-300 instrument by using a FactorFour™ (Varian VF-1 ms) dimethylpolysiloxane capillary column that had a 15 m length and 0.25 mm of i. d. Octane was used as the internal standard. Five reaction products were identified: cyclohexene epoxide, cyclohexanediol, 2-cyclohexen-1-ol, 2-cyclohexen-1-one and 2-cyclohexenyl hydroperoxide. Cyclohexene conversion was calculated from the yields of these five products.

### 2.5. Characterization Techniques

Powder X-ray diffraction was carried out using a PANalytical X'pert Pro Instrument (Cu Kα radiation). The gold content in the solid was determined by inductively coupled plasma spectrometry (ICP-OES) with an ICP PlasmaQuant PQ 9000 Analytic Jena spectrometer. Thermogravimetric analyses (TGA) were performed in a Perkin-Elmer TGA7 instrument, in air flow (40 mL/min) and a 20°/min heating rate from room temperature to 900 °C. The CHNS elemental analyses were performed in a LECO CHNS-932 instrument. Nitrogen adsorption–desorption isotherms were measured in a Micromeritics ASAP 2420 apparatus at the temperature of liquid $N_2$ ($-196$ °C). The samples were degassed in situ at 70 °C in vacuum for 16 h prior to analysis. The surface area was determined using the BET method, and the pore volume and the average pore size were calculated by the BJH method applied to the adsorption branch of the isotherm. The diffuse reflectance UV-vis spectra were recorded on a Cary 500 Varian spectrophotometer equipped with an integrating sphere with the synthetic polymer Spectralon as a reference. The data were expressed in absorbance units. Transmission electron microscopy (TEM) analyses were carried out in an FEI Titan XFEG (Thermo Fisher Scientific, Waltham, MA, USA) operated at 300 kV. The microscope was equipped with a CEOS spherical aberration corrector for the electron probe to assure a maximum spatial resolution of 0.8 Å and a (Oxford instruments, Abingdon, UK) silicon drift detector (SDD) for the energy-dispersive spectroscopy (EDS). The experiments were performed in the scanning mode using a high-angle annular dark field (HAADF) detector. X-ray photoelectron spectra (XPS) were collected using an SPECS instrument (SPECS Surface Nano Analysis GmbH, Berlin, Germany) with a UHV system (pressure

in the range of $10^{-10}$ mbar), equipped with a PHOIBOS 150 9MCD energy analyzer. A nonmonochromatic Mg Kα (1253.6 eV) X-ray source was used, working at a power of 200 W, with an acceleration voltage of 12 kV. The high-resolution regions were recorded with a pass energy of 20 eV. For the analysis, the powder samples were pressed into self-supporting wafers and stuck on the sample holder with double-sided adhesive conductive carbon tape. The spectra were referenced against the O 1s emission line assigned to silica (BE set to 532.9 eV) to correct for the charging effect. CasaXPS software (v2.3.16) was used for the quantification and fitting of the XPS spectra. Experimental peaks were deconvoluted into components (70% Gaussian and 30% Lorentzian), using a nonlinear, least squares fitting algorithm and a Shirley baseline. The relative atom ratios were calculated from the sum area of the core-level components using relative sensitivity factors provided by CasaXPS software (v2.3.16).

## 3. Results and Discussion

### 3.1. Characterization of the Materials

3.1.1. SBA-15 Functionalized with Thiol Groups

Thermogravimetric analysis (TGA) of the functionalized sample, SBA-SH (Figure 1), shows a weight loss at T < 140 °C centered at ~50 and 100 °C, attributed, respectively, to the desorption of water and some volatile organic material. This was followed by a sharp weight loss centered at ~340 °C, not present in the calcined sample (SBA-15c), which was assigned to the desorption/combustion of propylthiol groups present in SBA-15 materials [15,24].

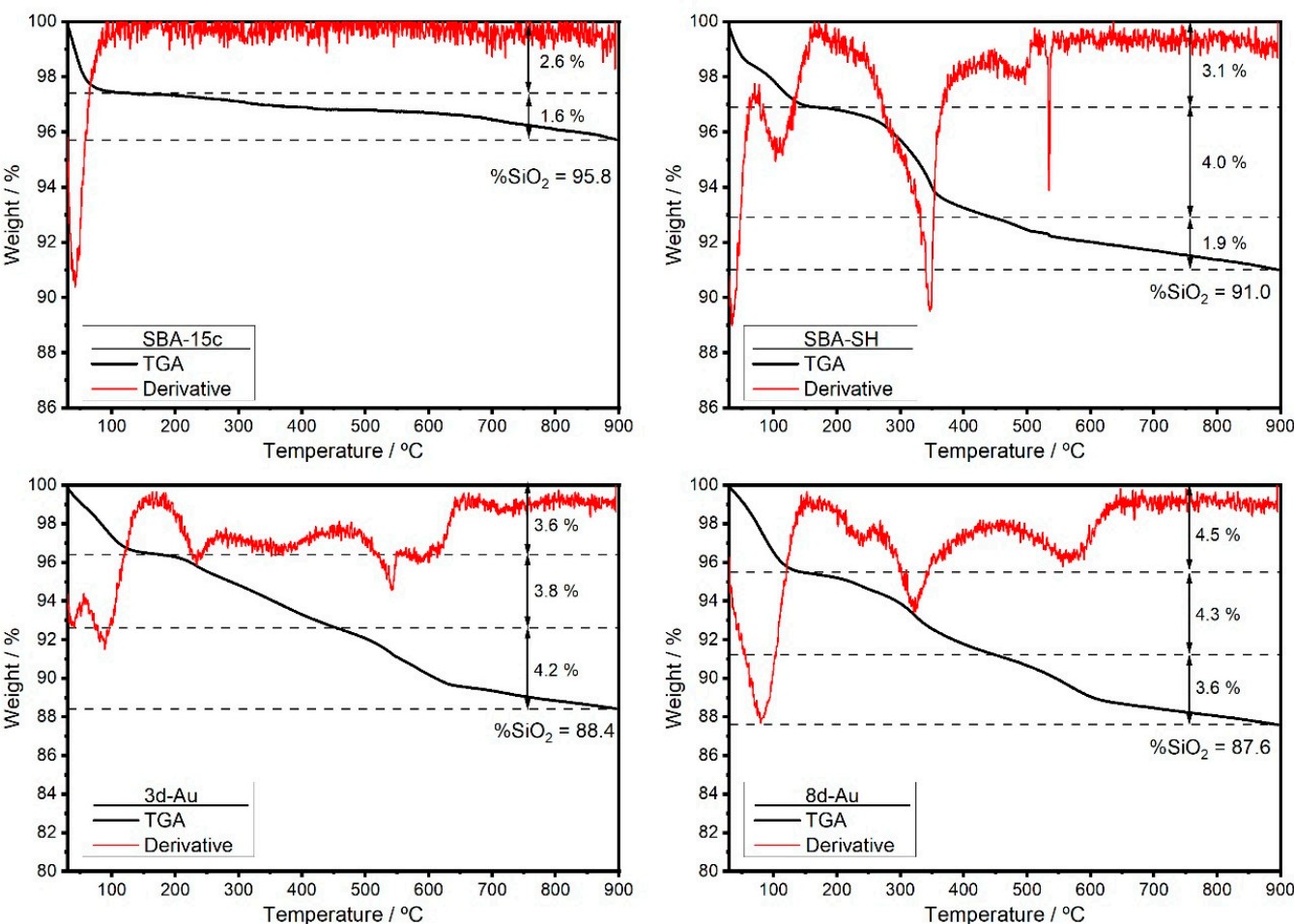

**Figure 1.** TGA (black line) and DTG (red line) of selected samples.

The chemical analysis (Table 1) confirms the effective incorporation of 1.33 wt.% of sulphur in the functionalized SBA-15 material, equivalent to a S/Si atomic ratio of 0.027. The C/S ratio (6.2) was higher than that corresponding to the thiol silane chain (C/S = 3). The excess of organic material could be attributed to methoxy groups that would remain attached to the support through a Si-O bond after they were removed from the starting trimethoxy silane during the functionalization process. Indeed, it has been shown [16] that hot ethanol easily reacts with the silanol groups of SBA-15 materials to form $Si-O-CH_2-CH_3$, and a similar reaction can take place here to produce $Si-O-CH_3$ groups. Indeed, the C/S equal to six agrees with this conclusion. The XRD evidences that the well-ordered structure of the calcined SBA-15 material remains unaffected by the functionalization (Figure 2), with the unit cell parameter unchanged (Table 2).

**Table 1.** Chemical composition and TGA results.

| Sample | Au (wt.%) | S (wt.%) | C (wt.%) | N (wt.%) | C/S (mol/mol) | S/Si (mol/mol) | TGA 170–400 °C (wt.% loss) | TGA 400–900 °C (wt.% loss) |
|---|---|---|---|---|---|---|---|---|
| SBA-SH | 0 | 1.33 | 3.11 | 0.03 | 6.3 | 0.027 | 4.0 | 1.9 |
| 3d-Au | 2.69 | 1.24 | 5.16 | 0.14 | 11.1 | 0.026 | 3.8 | 4.2 |
| 3d-Aux2 | 1.45 | 1.25 | 4.78 | 0.11 | 10.2 | 0.027 | 4.4 | 3.6 |
| 8d-Au | 0.31 | 1.25 | 5.31 | 0.13 | 11.4 | 0.027 | 4.3 | 3.6 |

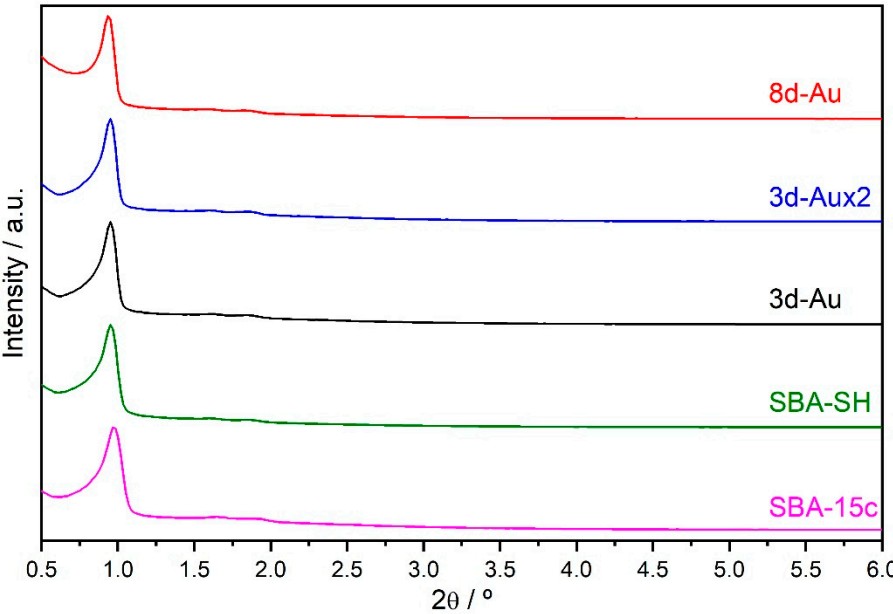

**Figure 2.** XRD patterns of the samples.

**Table 2.** Structural and textural properties of the samples.

| Sample | *d*-Spacing $d_{100}$ (nm) | Unit Cell Parameter $a_0$ (nm) | BET Surface Area (m²/g) | Total Pore Volume (cm³/g) | Micropore Volume (cm³/g) | Mesopore Volume (cm³/g) | Mesopore Size (nm) |
|---|---|---|---|---|---|---|---|
| SBA-15c | 9.275 | 10.7 | 742 | 0.79 | 0.09 | 0.70 | 8.7 |
| SBA-SH | 9.278 | 10.7 | 689 | 0.74 | 0.06 | 0.68 | 8.3 |
| 3d-Au | 9.297 | 10.7 | 587 | 0.63 | 0.05 | 0.58 | 8.1 |
| 3d-Aux2 | 9.304 | 10.7 | 606 | 0.66 | 0.05 | 0.61 | 8.1 |
| 8d-Au | 9.411 | 10.9 | 614 | 0.66 | 0.05 | 0.61 | 8.2 |

The incorporation of sulphur slightly decreased the surface area and the mesopore volume (Table 2), while the observed decrease in the average pore diameter from 8.7 nm

to 8.3 nm can be attributed to the presence of the mercaptopropyl moieties decorating the walls of the mesopores. This reduction in the pore size is the first suggestion, albeit indirect, that the functionalization affects the whole internal structure of the SBA-15. This conclusion is confirmed by the XPS results (Figure 3).

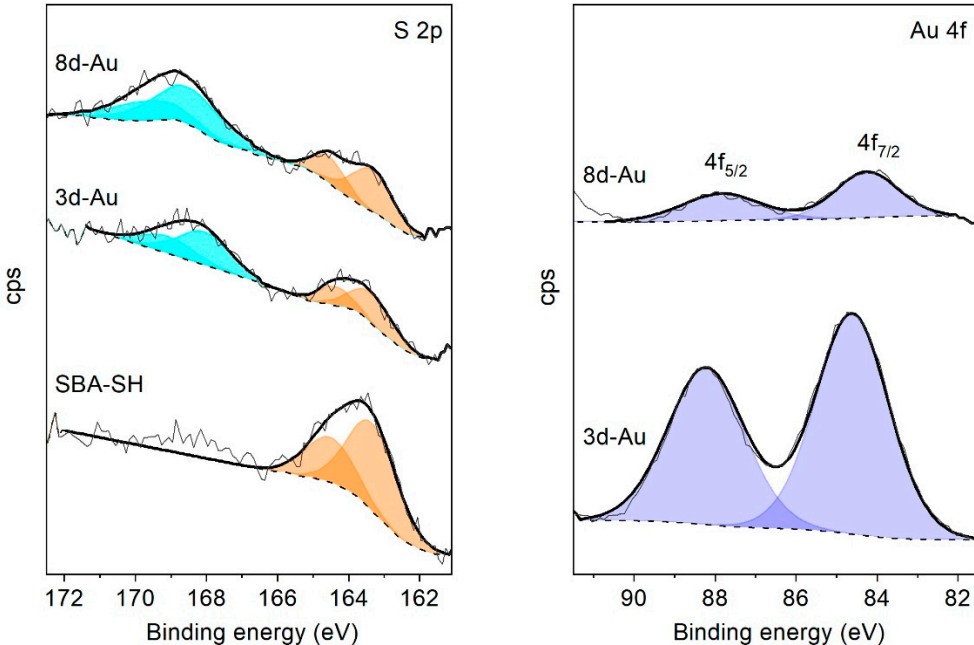

**Figure 3.** S 2p and Au 4f core-level XPS spectra. Signal intensities were normalized to a constant Si 2p peak area.

The S 2p core level spectrum of sample SBA-SH shows a broad asymmetric peak contributed to by the S $2p_{3/2}$ and S $2p_{1/2}$ components with a 2:1 intensity ratio and binding energies (BEs) of 163.3 eV and 164.5 eV (Table 3) characteristic of SH groups [25]. The S/Si ratio was 0.008, which is lower than that of the bulk material, 0.027. This result suggests that the more external rim of the SBA-15 particles is depleted in S compared with the more inner part of the material, the S/Si ratio of which should be close to that determined by bulk analysis. This is, therefore, confirmation that the interior of the SBA-15 particles was effectively functionalized by the mercaptopropyl groups.

**Table 3.** XPS core-level binding energies and atomic ratios.

| Sample | BE (eV) | | | | | Surface Atomic Ratio | | | | Bulk Atomic Ratio | | |
|---|---|---|---|---|---|---|---|---|---|---|---|---|
| | Si 2p | S $2p_{3/2}$ | S $2p_{1/2}$ | Au $4f_{7/2}$ | Au $4f_{5/2}$ | S/Si | Au/Si | Au/S | $S_{ox}$/S [a] | S/Si | Au/Si | Au/S |
| SBA-SH | 103.4 | 163.3 | 164.5 | - | - | 0.008 | - | - | 0 | 0.027 | 0 | 0 |
| 3d-Au | 103.5 | 163.3 168.5 | 164.5 169.7 | 84.6 | 88.2 | 0.007 | 0.007 | 1.11 | 0.58 | 0.026 | 0.009 | 0.356 |
| 8d-Au | 103.5 | 163.3 168.5 | 164.5 169.7 | 84.9 | 88.3 | 0.008 | 0.001 | 0.15 | 0.61 | 0.027 | 0.011 | 0.041 |

[a] Fraction of oxidized sulphur species.

### 3.1.2. Gold-Containing Catalysts

The SBA-15 functionalized material when contacted with the aliquots of the rosemary oil solution, as described in the Section 2, incorporated gold, the content of which was a function of the variables involved in the process (Table 1). Thus, it was higher for the sample prepared from rosemary oil that had been in contact with the gold aqua regia

solution for three days (sample 3d-Au, 2.69 wt.% of gold) but decreased as the contact time between the rosemary oil and the aqua regia solution was prolonged up to 8 days (sample 8d-Au, 0.31 wt.% of Au). In addition, if the rosemary/SBA-15 weight ratio duplicated (sample 3d-Aux2), the amount of gold immobilized decreased to 1.45 wt.%. Although it would be expected that the amount of Au immobilized increased for this sample, the higher concentration of the organic compounds present in the ethanolic solution contacted with the support, compared with that of sample 3d-Au, could disturb the accessibility of gold species toward the sulphur-containing anchoring groups. The structure of the Au/SBA-15 materials was not affected by the immobilization of gold (Figure 1) and neither was the unit cell parameter (Table 2). The surface area and pore volume decreased compared with the Au-free support, but the surface area was still approximately 600 m$^2$/g, while the average pore diameter was approximately 8 nm. The sulphur content remained practically unchanged (Table 1), but the Au samples were slightly richer in carbon (by ~2 wt.%) and hydrogen than the Au-free material, which suggests, together with the presence of traces of nitrogen (0.1–0.14 wt.%), that a small amount of organic material was incorporated during the treatment of the support with the Au-containing rosemary solution in ethanol. Though the sulphur content remained unaltered, the XPS spectra of the two samples examined, 3d-Au and 8d-Au, evidenced in both cases that the immobilization process modified the chemical state of a substantial fraction of sulphur present in the solid. In addition to the S 2p signal previously discussed and assigned to -SH groups, another spectral feature was clearly observed at approximately 169 eV, which can be split into two components at a BE of 168.5 eV and 169.7 eV, assigned to the S 2p$_{3/2}$ and S 2p$_{1/2}$ core-level peaks of sulphonic [25] or sulphonate groups -SO$_3^-$ [26], respectively. The oxidized sulphur species represent approximately 60% of the total sulphur, with almost no difference between both samples (Table 3). This partial oxidation of the sulphur during the gold immobilization process has been systematically observed previously by us upon using both rosemary [18] and eucalyptus oils [16]. This oxidation ability of the essential oils in prolonged contact with the aqua regia solution of gold has been attributed [18] to changes in the chemical nature of the terpenic compounds present in the essential oil, which are manifested by a notorious alteration of the oil color, which, being initially colorless, soon acquired a deep brown–red hue after three days, becoming clearly red after eight days.

The aggregation state of the gold in the materials was assessed by UV-vis spectroscopy and HRSTEM. The UV-vis spectra of the three Au/SBA-15 materials (Figure 4) showed an onset of the absorption at approximately 400 nm, which was more pronounced for the sample with the highest gold content (3d-Au), and a band at ~360 nm was observed, the intensity of which increased with the gold content. Another broad band at 250 nm–270 nm was also observed. However, no band of the surface plasmon resonance (SPR) usually centered at approximately 520 nm characteristic of AuNPs was found, which evidences that no gold nanoparticles were present in the catalysts. Spectral features in the range 250–450 nm have been reported in the UV-vis spectra of gold nanoclusters [27–30], which suggests that small gold entities at a nanocluster size (d < ~2 nm) were responsible for the spectral characteristics of these Au/SBA-15 samples.

Spherical aberration-corrected (Cs-corrected) STEM-HAADF analyses were carried out in order to obtain further insight into the nature of the gold entities present in the materials. Figure 5 shows images corresponding to samples 3d-Au and 8d-Au. The low magnification data (Figure 5a,b) present two different particles of sample 8d-Au, with the pores oriented parallel and perpendicular to the electron beam. The fast Fourier transforms (FFT) shown in the insets reveal an excellent pore ordering of the SBA-15 support, and were indexed within the *p6mm* space group obtaining unit cell parameters of a = b = 11.54 nm and c = 9.09 nm. The same type of data were obtained for sample 3d-Au in terms of the pore ordering and unit cell. For both samples, a closer observation is depicted in Figure 5c–f along the [001] zone axis of the SBA-15 support. In Figure 5c,d, corresponding to sample 3d-Au, the strong contrast observed belongs to the Au clusters (indicated by red arrows in Figure 5d), with a diameter that varies between 1 and 2 nm, which are homogenously distributed along

the entire SBA-15 particle. On the other hand, such a strong contrast was not observed for sample 8d-Au. For this sample, the low magnification image (Figure 5e) indicates a lower Au content along the support framework as compared to sample 3d-Au, in agreement with the bulk chemical analysis. A closer look at very high-magnification (Figure 5f) reveals the existence of small clusters in sample 8d-Au, in lower quantity, approximately 1 nm in size (red arrow) but also the presence of isolated Au atoms (yellow arrows). The same results were obtained when both samples were analyzed with the SBA-15 channels perpendicular to the electron beam (Figure 5g,h). While the sample 3d-Au contained numerous metal clusters (Figure 5g), sample 8d-Au exhibited "cleaner" surfaces, where single Au atoms can be observed. In both cases, a closer observation of the clusters and of the atoms are depicted in the inset of each image. Au NPs were not observed in any of the catalyst particles analyzed. These results are in agreement with the above discussed UV-vis spectra of these samples. The chemical composition was obtained through energy-dispersive X-ray spectroscopy (EDS), as shown in Figure 5i, where the Si and O signals are owed to the SBA-15 framework, and the S and Au were homogeneously distributed along the entire particle.

X-ray photoelectron spectroscopy allowed for obtaining further insight into the chemical state of these small gold entities. The results of samples SBA-HS, 3d-Au and 8d-Au are depicted in Figure 3 and collected in Table 3. The Au 4f core-level XPS spectrum of sample 3d-Au (Figure 3) shows two broad and symmetrical signals of the Au $4f_{7/2}$ and Au $4f_{5/2}$ components at a BE of 84.6 eV and 88.2 eV, respectively. The spectrum of the sample with a lower gold content, 8d-Au, was qualitatively similar, though of lower intensity, with the Au $4f_{7/2}$ and Au $4f_{5/2}$ signals appearing at a slightly higher BE, 84.9 eV and 88.3 eV. The BE of these signals was therefore shifted by +0.6–0.9 eV toward higher energy than that corresponding to bulk Au(0), which was 83.9 ± 0.1 eV for Au $4f_{7/2}$ [31]. It has been reported that the BE of Au nanoclusters increases as the cluster size decreases [31–33]. Therefore, the observed high-energy shift of the Au 4f core-level observed in our samples is consistent with the presence of highly dispersed gold atoms forming part of small nanoclusters and present even as isolated species in agreement with the electron microscopy observation.

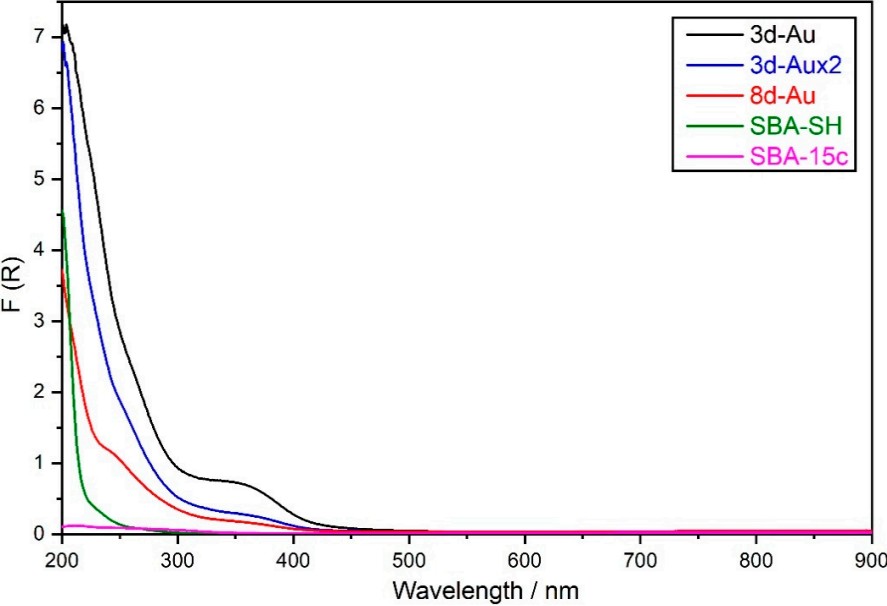

**Figure 4.** UV-vis spectra of the samples.

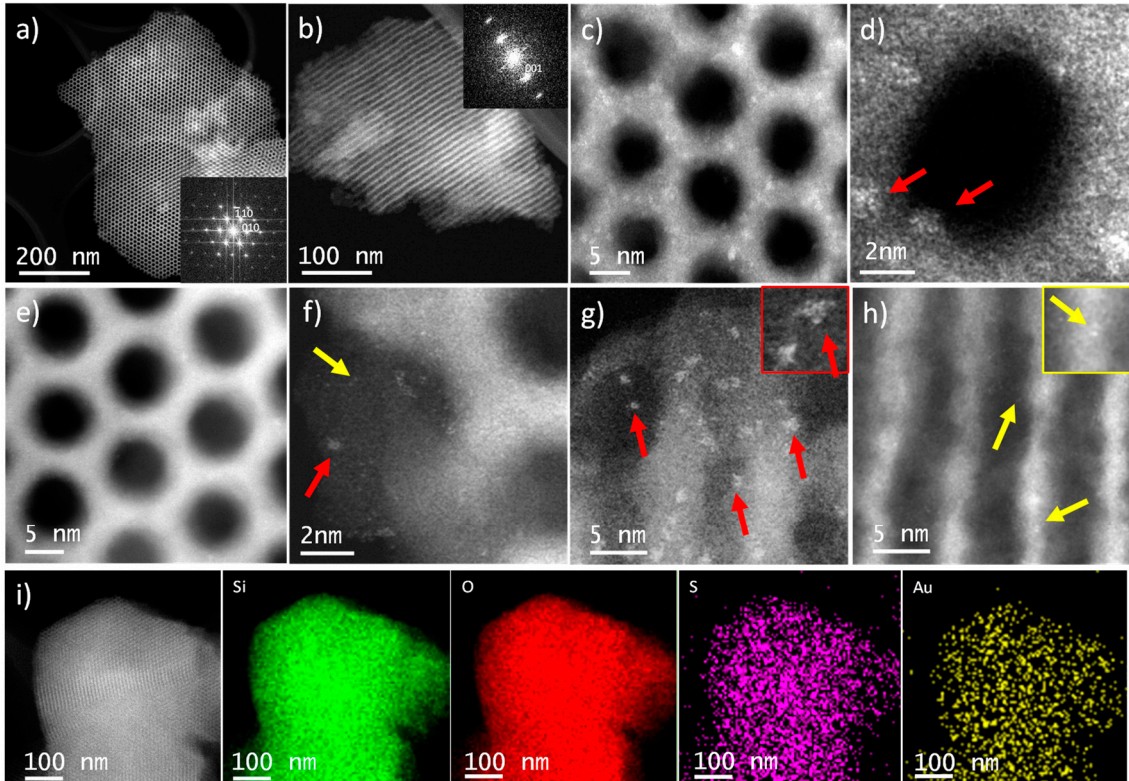

**Figure 5.** Cs-corrected STEM-HAADF analysis of the 3d-Au and 8d-Au materials. Low-magnification images of sample 8d-Au with (**a**) the channels of SBA-15 support parallel to the electron beam (FFT shown in inset) and (**b**) channels perpendicular to the electron beam (FFT in inset). A closer observation is shown in (**e**) and high-magnification images observing the SBA-15 channels parallel (**f**) and perpendicular to the electron beam (**h**); a closer observation of the single atoms is shown in the inset (**h**). Low-magnification images of sample 3d-Au are shown in (**c,d**), as well as a high-magnification image in (**g**), with a closer observation of the clusters shown in the inset. Color code: red arrows point to the Au clusters, while the yellow arrows point to the Au single atoms. (**i**) EDS chemical analysis showing the ADF image where the analysis was carried out and the chemical composition maps of each element.

It is also interesting to compare the Au distribution at the surface (as determined by XPS analyses) and bulk, collected in Table 3. The surface Au/Si ratio was only slightly lower than that of bulk, but owing to the strong sulphur depletion at the surface, the Au/S ratio was nearly three times higher at the surface than in bulk, being 1.1 for sample 3d-Au. This suggests that, on average, at the surface of the sample every atom of gold would be interacting with one sulphur atom. However, the TEM studies discussed above reveal that gold clusters were present in the sample, and hence, a relatively large fraction of the Au atoms would just be buried inside the metal particle core, unable to interact with sulphur groups. Moreover, it should be noticed that the BE of the S atoms of the thiol groups in the functionalized material did not change upon gold immobilization (Table 3), and it has been reported that the BE of thiol groups shifts by nearly 1 eV to lower energy upon interaction with gold (S $2p_{3/2}$ = 162.0 eV) [31]. As this shift was not observed in our samples, this probably means that the Au atoms eventually interacting with sulphur at the surface involved such a small fraction of the total sulphur present in the sample, that it was indiscernible by the XPS instrument. Additionally, it could also be possible that the main interaction between the gold and sulphur species took place mainly through sulphonic and not through thiol groups, a possibility that agrees with the observed catalytic activity, as is discussed later. However, the BE of the $-SO_3^-$ groups was also not affected by the immobilization of gold, but, again, the fraction of the total sulphonic groups affected by

such interaction and its surface concentration would be so small that its detection would be beyond the resolution power and sensitivity of our XPS instrument. Taking into account all these considerations, it is possible that a large fraction of the sulphur groups of the support would not be involved in chemical interactions with gold.

### 3.2. Catalytic Tests

3.2.1. Tests at Ambient Lighting in Presence of TBHP

The three Au-containing catalysts previously described were tested in the oxidation of cyclohexene with molecular oxygen in the conditions described in Section 2. The reaction setup was placed inside a fume cupboard illuminated by fluorescent tubes and, hence, exposed to the ambient lighting therein prevailing. The corresponding cyclohexene conversions as a function of reaction time are depicted in Figure 6, together with the results of the Au-free SBA-SH sample included for comparison purposes.

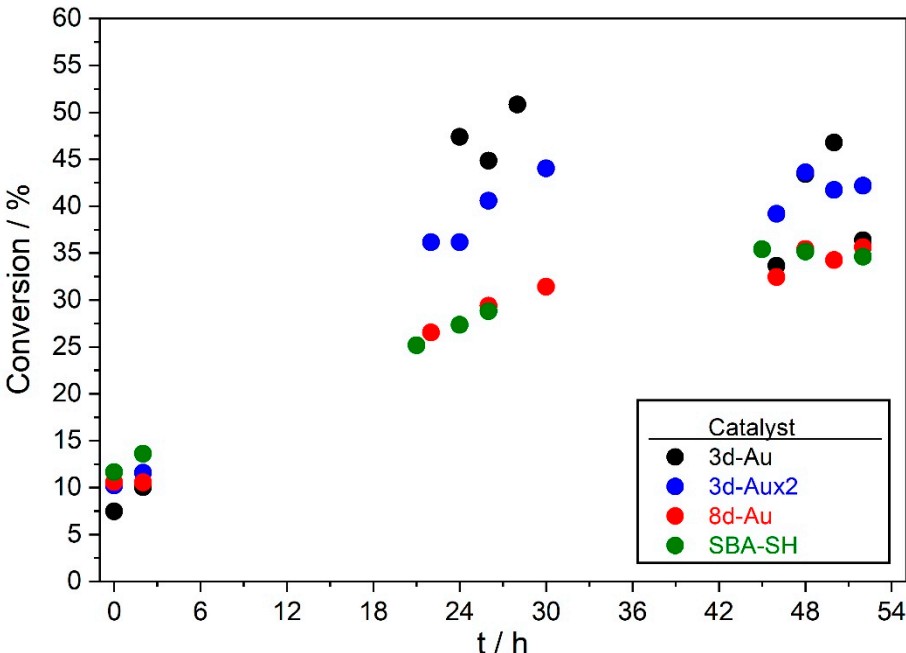

**Figure 6.** Cyclohexene conversion as a function of the reaction time for the catalytic experiments carried out at ambient lighting in the presence of TBHP. The experimental conditions are described in Section 2.4 of the Materials and Methods section.

It can be observed in Figure 6 that the conversion increased with the gold content of the catalysts, being ~47% for 3d-Au (2.69% of Au) and ~37% for 3d-Aux2 (1.45 wt.% of Au) after 24 h of reaction. The catalyst prepared from rosemary oil that had been in contact with the aqua regia gold solution for 8 days (8d-Au) contained a very low amount of gold (0.31 wt.%), and its catalytic activity was only slightly above that of the Au-free SBA-SH sample. It is also noteworthy that the conversion increased up to nearly 24–28 h of reaction time, to remain nearly constant after that. The reason for that loss of catalyst activity beyond one day of reaction time resides in the changes experienced by the catalysts in the reaction medium. It has been observed that the catalysts, being nearly white as prepared, become purple colored during the reaction, which is a visual indication that suggests the presence of AuNPs. This was indeed confirmed by the UV-vis spectra of the used catalysts, where the surface plasmon resonance (SPR) band characteristic of AuNPs was observed, while the bands associated to AuNCs visible in the fresh catalysts vanished (Figure 7).

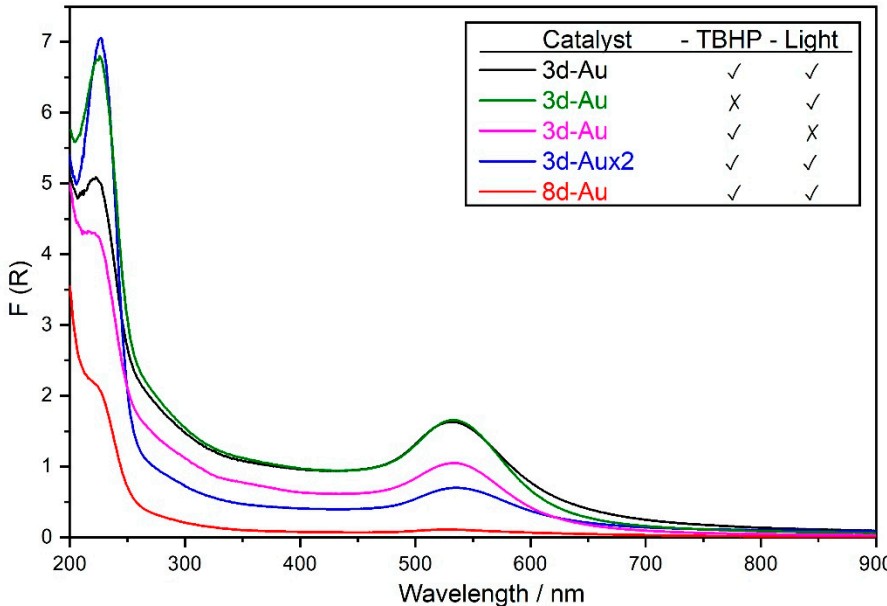

**Figure 7.** UV-vis spectra of the used catalysts.

The SPR band is asymmetric and broader on the high wavelength side, and its maximum absorbance is in the range 531 nm (for 8d-Au) to 539 nm (for 3d-Aux2), being 534 nm for 3d-Au. These values correspond to AuNPs of sizes between ~50 nm and ~65 nm [34]. Therefore, the AuNCs and the isolated gold atoms initially present in the fresh catalysts aggregated in the reaction medium to form large AuNPs, the average activity of which would be much lower than that of the smaller gold entities formerly present in the catalysts, if only because the rapid decrease in the gold surface atoms as the particle size increases. Indeed, gold entities with a size < 2 nm are generally more active than AuNPs (sizes > 2 nm) in many reactions [35]. In this regard, it has been reported that gold entities with a size ~1.4 nm derived from 55-atom gold clusters are active catalysts for the oxidation of styrene with $O_2$ in the absence of any radical initiator, but particles with a size > 2 nm were inactive for this reaction [36]. This aggregation process of small gold entities has previously been systematically observed by us in Au catalysts prepared by the two-liquid phase method here described, based on S-containing SBA-15 materials [15,16,18], and it has also been reported for triphenylphosphine-stabilized Au clusters immobilized on $SiO_2$ during the oxidation of cyclohexene [37]. In this last study, the catalysts' activity was associated to the formation of particles > 2 nm.

The activity of the catalysts studied here can be compared with that of a catalyst reported in [18], prepared also from plate-like SBA-15 materials but functionalized with thiol groups by in situ procedures, incorporated during the synthesis. The gold content of this catalyst was 1.09 wt.%, and owing to the disparity in the metal content among the several samples, their activity should be compared based on TON values (moles of cyclohexene converted per mol of Au atom), in this case at 24 h of reaction time. The TON values for the 3d-Au and 3d-Aux2 catalysts were $3.3 \times 10^3$ and $4.9 \times 10^3$, respectively, while that of [18] was $6.5 \times 10^3$. However, the superior activity exhibited by the last catalyst was not the only difference among them, because it kept converting cyclohexene at a steady rate up to at least two days to reach ~60% conversion [18]. Its sulphur content, 1.67%, was close to that of the two other catalysts, and the XPS spectrum showed a nearly similar fraction of oxidized sulphur at the surface, 57% of the total. The main difference between the two types of catalysts that might contribute to explaining the observed differences in the catalytic performances seems to reside in the Au/S ratios, both at the surface and in the bulk. The surface Au/S ratio of sample 3d-Au (Au content = 2.69 wt.%) was 1.1, while the surface Au/S ratio of the sample reported in [18] (Au = 1.09 wt.%) was 0.067, more than one order of magnitude smaller. Moreover, the bulk Au/S ratio was 0.35 in the former

(Table 3) and 0.10 in the latter. These differences in the corresponding Au/S ratios could be relevant concerning the catalytic behavior of the samples, as it can be expected that their performance was influenced by the chemical interaction established between the Au species (AuNCs and isolated Au atoms) and the sulphur atoms of the support. It could be expected that a relatively higher proportion of S atoms to Au atoms would slow down the rate of the aggregation process, which nevertheless also occurs in the sample reported in [18]. However, with the current knowledge gained on this complex catalytic system, an aspect of which is the transient nature of the gold species, it is not possible to discern the precise nature of the gold–sulphur interaction at the molecular scale. Nevertheless, it has been shown that the cyclohexene oxidation activity of the Au/SBA-15 samples containing only thiol groups was extremely low but increased enormously if part or all of these groups were oxidized to sulphonic species [15–18]. This suggests that -$SO_3^-$ groups are, indeed, involved in the interaction with gold species, though it does not necessarily exclude the simultaneous participation of some –SH groups.

Selectivity to the different reaction products as a function of conversion is depicted in Figure 8. The three products resulting from the allylic oxidation of cyclohexene, 2-cyclohexen-1-one, 2-cyclohexene-1-ol and 2-cyclohexenyl hydroperoxide, were dominant, and at high conversion they accounted for over 85% of the total products. At low conversion, ~10%, the selectivity of these three products was similar, close to ~30%, but then the selectivity to the enol decreased to stabilize at ~20–25% for conversions higher than ~35%. The selectivity to the hydroperoxide decreased more rapidly with conversion to reach values of 10–15% at high conversion. Contrary to these two products, the selectivity to the enone steadily increased with conversion to become the major product, with enone/enol ratios of ~2. The catalysts were also active for the epoxidation of the double bond, though in a lower extension than the activity shown for allylic oxidation. The selectivity to the cyclohexene epoxide decreased with conversion, while at the same time 1,2-cyclohexanediol began to be formed, which suggests it comes from the hydrolysis of the epoxide.

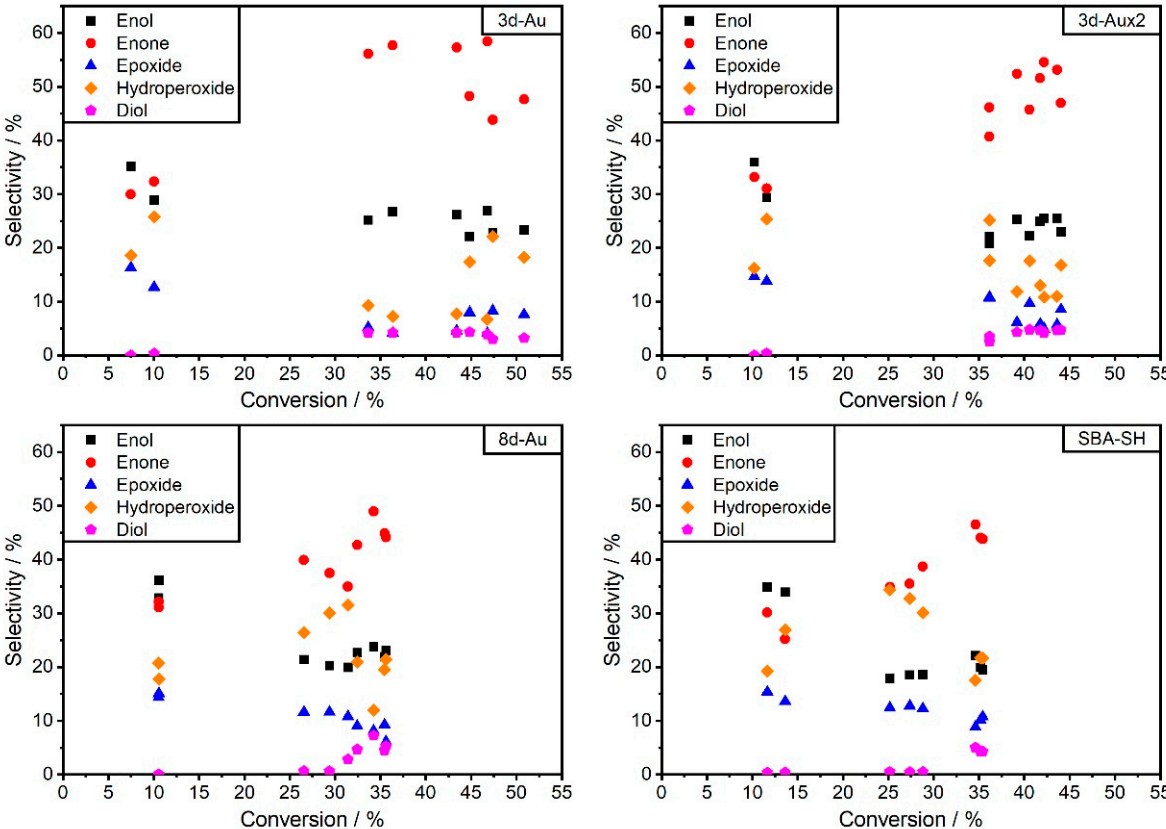

**Figure 8.** Selectivity as a function of conversion.

### 3.2.2. Tests Either in Darkness or in Absence of TBHP

Catalytic tests by using the sample 3d-Au were also carried out in darkness, by housing the reaction setup in an on-purpose built cardboard box and also in a separate experiment without adding the TBHP initiator to the reaction mixture but with exposure to ambient lighting. The conversions as a function of the reaction time are depicted in Figure 9 and the corresponding selectivities in Figure 10.

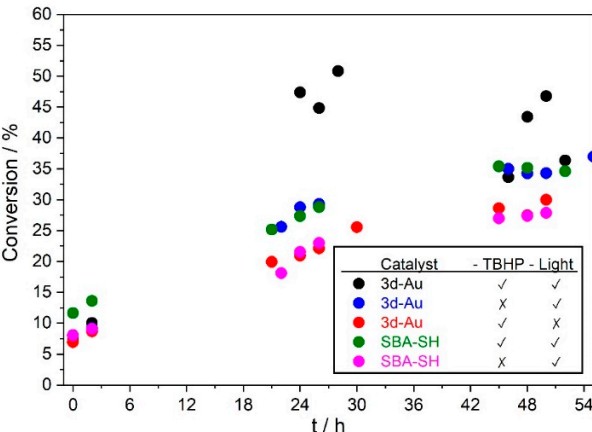

**Figure 9.** Comparison of the conversion of cyclohexene as a function of time of the catalyst 3d-Au carried out under different reaction conditions.

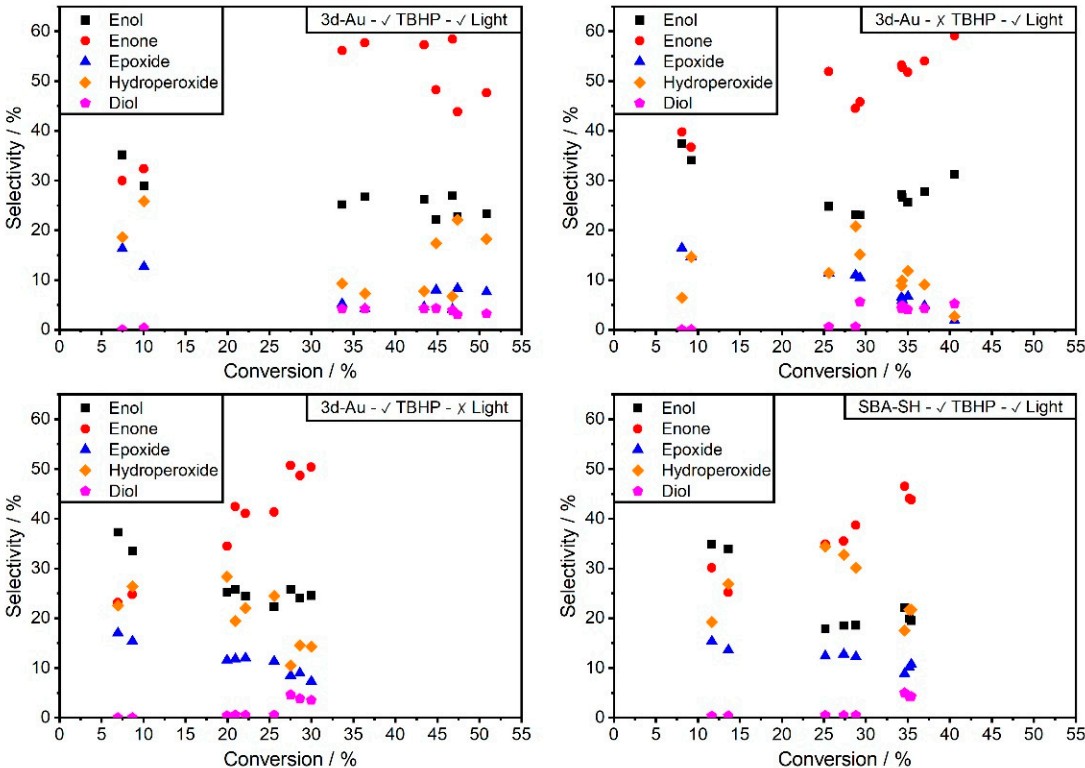

**Figure 10.** Selectivity to the different reaction products for catalyst 3d-Au for the reactions displayed in Figure 9.

If the reaction is carried out in darkness, the activity of the gold is virtually completely suppressed, as the conversion drops to the level found for the Au-free SBA-SH used as reference (blank reaction). If no hydroperoxide initiator (TBHP) is added, the activity also drops to a level only slightly above that of the Au-free catalyst, the activity of which also decreases in the absence of TBHP. On the other hand, no major influence of these reaction

conditions on the selectivity to the different reaction products is found. The necessity of adding this peroxide to the reaction mixture is linked to the use of commercial cyclohexene in which a stabilizer (i.e., an inhibitor of cyclohexene autoxidation) is present (this is the type of cyclohexene used in this work, stabilized with butylhydroxytoluene). The formation of peroxide radicals is a required step in the oxidation of cyclohexene by $O_2$ [37], and it has been reported that if a stabilizer-containing cyclohexene is used to oxidize this compound with $O_2$ by using Au clusters immobilized on nonfunctionalized all-silica SBA-15, no cyclohexene conversion was observed in the absence of a peroxy initiator [37]. It has also been reported that the oxidation of cyclohexene containing stabilizers (scavengers of free radicals) requires the use of TBHP as an initiator [23], but initiator-free oxidation of this and other cyclic olefins (from C5 to C8) with $O_2$ using carbon-supported AuNPs is possible by using stabilizer-free olefins [38]. In this way, the basic role of the peroxide initiator (TBHP in this work) is to counteract the stabilizer [38]. It is also interesting to notice that in our case, in the TBHP-free system, it was still observed the formation of AuNPs from the aggregation of the AuNCs and isolated Au atoms initially present in the fresh catalyst (Figure 6). This aggregation process of gold, therefore, occurs within the reaction mixture as a function of time independently on the gold activity. Contrary to this finding, it has been reported that AuNCs do not evolve toward AuNPs in gold supported on SBA-15 if the oxidation reaction is suppressed using cyclohexene containing a stabilizer in the absence of any peroxide initiator [37].

Concerning the influence of light, the observed total suppression of the gold activity by carrying out the reaction in darkness clearly points to a photo-assisted oxidation of cyclohexene by $O_2$ in the conditions prevalent in this work. It is worth noting in this regard that the mesoporous silica SBA-15 is basically an insulating material, rich in hydroxyl groups, however, and it is hence not expected to participate directly in electron transfer processes and, hence, would not contribute significantly to light-activated chemical reactions. The influence of ambient light in this reaction would eventually lead to re-evaluating literature reports on olefins oxidations (and eventually other substrates, such as alcohols, as well) by $O_2$ triggered by gold catalysts, whenever the catalytic tests at atmospheric pressure in conventional glass flasks not intentionally protected against ambient lighting were carried out. We wonder how often reported heterogeneous gold catalysis should actually be (inadvertently but largely) photo-assisted. As previously discussed for the reaction carried out in the absence of TBHP, also in the absence of light the gold entities, initially small (d < 2 nm) were inactive, but nevertheless they still evolved toward AuNPs. The formation of AuNPs within the reaction medium can in this way lead to plasmonic catalysis. It has been reported in this regard that a series of AuNPs supported on mesoporous silica spheres (MS), and on the ordered mesoporous materials KIT-6, SBA-15 and MCM-41, showed, under visible-light illumination, the optimum conversion in the oxidation of glycerol with $O_2$ for AuNPs with a diameter of 2 nm. Moreover, no influence of illumination was found for the catalysts containing isolated gold atoms and/or clusters with ~1 nm size [39]. As the formation of AuNPs was, indeed, observed in our system, it is most probable that these gold entities with sizes $\geq$ 2 nm were responsible for the observed photocatalytic activity. However, additional work is required to determine the nature of the gold entities that would be light-active for cyclohexene oxidation under the reaction conditions here reported.

## 4. Conclusions

Plate-like all-silica SBA-15 mesoporous material was successfully functionalized using a post-synthesis method with mercaptopropyl groups without distorting the pore hexagonal ordering of the parent support. The sulphur groups were distributed on the whole particles, with the surface being depleted of this element. The gold species produced in a two-liquid phase system were immobilized on the S-containing support as nanoclusters and isolated gold atoms, which were nearly homogeneously distributed across the support, with the surface only slightly depleted of gold. The immobilization process preserves the structural ordering and porosity of the S-bearing support but produces the oxidation

of nearly 60% of the thiol groups to sulphonic groups. These materials are active in the oxidation of cyclohexene with molecular oxygen, and it has been found that the gold entities initially present in the catalysts evolved to form large AuNPs during the reaction. This transient nature of the gold species deeply affects the performance of the catalysts. The observed lower catalytic activity of these materials compared with that of the related samples in which the mercaptopropyl groups were incorporated during synthesis was attributed to this gold aggregation process, which is probably regulated by the Au/S ratio of the catalysts: a high ratio would lead to an excessively fast growing of the gold nanospecies, being the activity of the AuNPs decreasing as their size increases. On the contrary, by increasing the population of sulphur atoms interacting with the gold species, the aggregation process would slow down, though not prevented. Indeed, factors such as the density of S-groups on the surface of the mesopores, their distribution homogeneity at nanoscale within the support and the thiol/sulphonic groups ratio, would all have an impact on the catalysts' performance.

It was found that the catalysts were inactive in darkness and, hence, the reactions carried out under exposure to ambient lighting were actually photo-assisted. The AuNPs formed in the reaction medium are responsible for this phenomenon of plasmonic catalysis. However, the transient nature of the gold entities present in the reaction medium would require additional studies to determine their relative contribution to activity during the whole reaction period. This finding adds a new dimension to the above noticed complex nature of these catalysts. Finally, the presence in the reaction medium of the radical initiator TBHP is required to obtain catalytic activity due to the presence of radical inhibitors in the cyclohexene source.

**Author Contributions:** Conceptualization, J.P.-P. and J.A.; Investigation, R.D., Á.M., C.M.-Á. and R.d.l.S.; Supervision, J.A., J.P.-P. and R.d.l.S.; Writing—original draft preparation, J.P.-P.; Writing—review and editing, J.P.-P., C.M.-Á., Á.M., R.D. and R.d.l.S.; J.A.; Project administration, J.P.-P.; Funding acquisition, J.P.-P., C.M.-Á., Á.M. and R.D. All authors have read and agreed to the published version of the manuscript.

**Funding:** This research was funded by MCIN/AEI/10.13039/501100011033, grant number: PID2019-107968RB-I00. R.S. acknowledges MCIN/AEI for a predoctoral grant (PID2019-107968RB-I00). A.M. acknowledges the Spanish Ministry of Science (RYC2018-024561-I), the regional government of Aragon (DGA E13_20R) and the National Natural Science Foundation of China (NSFC-21835002). R.D. acknowledges Sociedad Española de Catálisis (SECAT) for a summer internship grant.

**Conflicts of Interest:** The authors declare no conflict of interest.

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
