# Peer review of "Gold Clusters Immobilized by Post-Synthesis Methods on Thiol-Containing SBA-15 Mesoporous Materials for the Aerobic Oxidation of Cyclohexene: Influence of Light and Hydroperoxide"

_chemistry, doi:10.3390/chemistry5010038_

Round 1
Reviewer 1 Report
A high-quality investigation of the cyclohexene oxidation, utilizing the required analytical techniques and bringing along new impulses for the field.
Overall, I enjoyed reading it.
Before a publication, some points need to be addressed:
"are subjected to increasing exploration."
-- here, at the end of the sentence, add the following key reviews of the field:
Chem. Rev. 2021, 121, 9113-9163. "Homogeneous and Heterogeneous Gold Catalysis for Materials Science"
Angew. Chem. Int. Ed. 2006, 45, 7896-7936. "Gold Catalysis"
"heated up to 500 ºC for 1h."
-- between a number and a unit there always must be a space. Thus "1 h" instead of "1h"
Please check in the whole manuscript (e.g. "after 24h of reaction.")
Catalytic conversions:
It seems that the authors did not check for leaching of gold (ICP would allow that, and they have used that for the determination of the gold contents in the catalyst characterization, so they hava access).
Does the supernatant solution show activity without the catalyst? Does it contain gold?
In this context, it is also important to mention, that soluble, homogneneous gold complexes show photochemically induced catalysis, see the recent review of Jin Xie:
Chem. Rev. 2021, 121, 8868-8925. "Light in Gold Catalysis"
Reviewer 2 Report
Herein the author presents the work on gold nanospecies produced by a historically-inspired two-liquid phase system that have been immobilized on plate-like mesoporous silica SBA-15 functionalized with mercaptopropyl groups by a post-synthesis method. The resulting materials were examined for the oxidation of cyclohexene with 14 molecular oxygen at atmospheric pressure. Overall, this paper is very well-written and well-organized. The is done with care and considering the importance of heterogeneous gold catalysis, the referee has no doubt that the work deserves publication. Hence, this referee strongly recommends the publication of the manuscript. Before publication, it is advisable to cite a review on heterogeneous gold catalysis. For instance, ChemCatChem. 2011, 3, 1121; Chem. Soc. Rev. 2008, 37, 2096 etc.
Reviewer 3 Report
In this paper, the authors compared the physicochemical properties and catalytic performance of these materials with those of previously reported related materials functionalized by in-situ methods, by which the gold species produced in a two-liquid phase system are immobilized on the S-containing support as nanoclusters and isolated gold atoms. It was found that the catalysts were inactive in darkness and the AuNPs formed in the reaction medium would be responsible for this phenomenon of plasmonic catalysis. Major revisions should be considered before acceptation.
1. How did the authors deconvolute the XPS spectra? What was the reference energy for all XPS spectra? It is impossible to talk about BE without providing the applied reference energy.
2. For Figure 5, please provide the element mapping with SETM images.
3. For the cyclohexene oxidation reaction with molecular oxygen, please provide the catalytic stability of the catalysts in details in the revised version. As much, the reaction conditions are suggested to add in the caption of Figure 6.
4. The characterization of spent catalysts after reactions is missed in the paper. Please provide the information about the structural stability of catalysts during the reactions.
Round 2
Reviewer 1 Report
I follow the authors' Argument that soluble nanoparticles are not involved.
Thus, publish.
Reviewer 3 Report
This work has been improved and might be accepted.